# Evaluating Family Planning Organizations Under China’s Two-Child Policy in Shandong Province

**DOI:** 10.3390/ijerph16122121

**Published:** 2019-06-14

**Authors:** Lizheng Xu, Fan Yang, Jingjie Sun, Stephen Nicholas, Jian Wang

**Affiliations:** 1Center for Health Economics Experiment and Public Policy, School of Public Health, Shandong University, Key Laboratory of Health Economics and Policy Research, NHFPC (Shandong University), Jinan 250012, China; sddxxlz@163.com (L.X.); 15153132375@163.com (F.Y.); 2School of Health Sciences, Wuhan University, Wuchang District, Wuhan 430071, China; 3Shandong Provincial Health and Family Planning Information Center, Jian 250014, China; sunjingjie163@163.com; 4School of Management and School of Economics, Tianjin Normal University, West Bin Shui Avenue, Tianjin 300074, China; 5Newcastle Business School, University of Newcastle, University Drive, Newcastle, NSW 2038, Australia; 6Guangdong Research Institute for International Strategies, Guangdong University of Foreign Studies, Baiyun, Guangzhou 510420, Guangdong, China; 7Top Education Institute, 1 Central Avenue, Australian Technology Park, Eveleigh, Sydney, NSW 2015, Australia; 8Dong Fureng Institute of Economic and Social Development, Wuhan University, Dongcheng District, Beijing 100010, China; 9Center for Health Economics and Management at School of Economics and Management, Wuhan University, Wuchang District, Wuhan 430072, China

**Keywords:** family planning, government financing, organizational change

## Abstract

Background: The 2015 two-child policy was the most important institutional change in China’s family planning since the 1978 one-child policy. To implement the two-child policy, China merged the former health departments and family planning departments into the new Health and Family Planning Commission organization. We collected and analyzed funding and expenditure data, providing a novel approach to assessing the family planning outcomes under China’s two-child policy. The paper shows how the management structure and funding levels and streams shifted with the new two-child policy and assesses the new management structure in terms of the ability to carry out tasks under the new family planning policy. Methods: We collected data on the funding, structure of expenditure and social compensation fee in Shandong province from 2011 to 2016, to evaluate how resources were allocated to family planning before and after the organizational change. We also collected interview data from family planning administrators. Results: While total family planning government financing was reduced after the organizational change, expenditures were shifted away from management to family planning work. Funding (80%) was allocated to the grass-root county and township levels, where family planning services were provided. The overlapping work practices, bureaucracy, and inefficiencies were curbed and information flows were improved. Conclusions: The new Health and Family Planning Commissions shifted resources to carry out the new family planning policy. The aims of the two-child policy to reduce inefficiencies, overlapping authorities and excessive management were achieved and expenditures on family planning work was enhanced and made more efficient.

## 1. Introduction

China’s family planning institutions have a long history, punctuated by significant policy shifts and changes in the organizations that implement population control. The 2015 institutional change from China’s one-child to two-child family policy was the most significant change in China’s family planning since the 1978 one-child policy. This paper asks the question: Did the management structure and funding levels and streams shift in response to the new two-child policy and did the new management structure promote the new family planning policy? 

In 2015, China legislated new constraints on families, replacing the law allowing families only one child, with the two-child policy, constraining families to two children. The one-child and two-child policies are formal institutions or humanly devised legal constraints that shape human interaction and behavior [1,2]. Formal institutions are the “rules of the game”, or officially legislated legal rules, which the state enacts and enforces. Individuals and family planning commissions are “players” or organizations, that follow (individual families) or implement (family planning commissions) the rules of the game [1,2]. With the institutional change to the two-child policy, new organizations, the Health and Family Planning Commissions (HFPCs), were established in 2013 to replace the old National Population and Family Planning Commission and to orchestrate the 2015 shift to the two-child policy [3]. One novel and powerful approach to evaluating China’s post-2015 two-child policy is to assess the changes in resource flows and work practices in the new HFPCs compared to the old National Population Department and the old Family Planning Department. Instituted at five levels of government (national, provincial, city, county, and township) to promote and implement the two-child policy, resource flows and work practices in the HFPCs directly determined the quality and quantity of family planning services, health services and population management [4]. Families are also organizations that responded to the new two-child legal environment. We are not assessing how individual families adjusted their family planning decisions to the two-child policy, which included factors such as personal preferences, family economic affluence, age at marriage, mortality rates, education levels and costs of children.

Institutional or legal changes to China’s family planning have been a mix of evolution, modifications and sharp policy changes. Between 1955 and 1978, family planning service organisations put into effect China’s changing legal rules on family planning, including launching nationwide birth control programs to address the economic and social consequences of rapid population growth [5]. In 1963, the State Council Family Planning Commission was formed to manage population control and coordinate city and province-level family planning commissions. While the work of the Family Planning Commission was disrupted by the Cultural Revolution, the State Council’s 1971 “Report on Family Planning Work” marked the emergence of China’s one-child policy, which legally determined how individual families planned their families and how family planning service organizations operated [6]. After the one-child policy was incorporated into China’s Constitution in 1978, a new family planning organization, the Population and Family Planning Commission, was created to plan, implement and guide the one-child policy. Over the next two decades, the government made a series of legal modifications to family planning policy, such as allowing rural families with only one girl and some minority nationalities to have two children [6,7]. These minor legal changes also saw changes in the family planning organizations, for example family planning related social organizations, enterprises and research organizations emerged, such as the China family planning association and the China population association [8]. Between 1991 and 2013, the one-child policy and family planning organizations remained relatively stable, with no significant changes, until the 2013 separate child policy, which allowed families to have two children if one parent was an only child [9], and the legislating of the two-child policy in 2015.

Under the one-child policy, China’s family planning organizations controlled population growth and contributed to China’s economic and social development, such as maximizing resources allocated to child rearing and, in spite of skewing the sex ratio at birth, improved some aspects of gender equality, especially education opportunities and resources allocated to girls [10,11]. But the one-child policy also had some negative effects, like accelerating population aging, decreasing the size of the working-age population and denying families choice over family size. Under these pressures, the existing family planning organizations were judged unsuited for the separate child and two-child-policy. The existing family planning organizations led to the decentralization of population work, which was divided among different departments, with unclear rights and responsibilities at the provincial and city level [12]. Some work, such as reproductive health and birth control surgery, was subject to overlapped jurisdictions by the family planning organizations and the health department. The result was untimely information communication, resulting in the duplication and waste of resources [9]. China’s family planning organizations were seen as organizationally unfit for the two-child family planning policies, enshrined in Chinese law in 2015. 

To adapt to the forthcoming two-child institutional change, China reformed its family planning, population control and health organizations. In 2013, the former Ministry of Health and National Population and Family Planning Commission were merged into National Health and Family Planning Commission, with comparable organizations at each level of government. Shandong province established its commissions in 2014. The purpose of the organizational change was to facilitate the implementation of the new two-child policy, reallocate funding and expenditure to family planning work, reduce overlapping work practices between departments, eliminate wasteful funding and streamline information flows and decision-making [12]. Did the new HFPCs achieve these goals? In other words, were the organization changes in management structure and funding levels promoting the new family planning policy?

Previous research on China’s family planning has mainly focused on family planning services [13,14], the evaluation of China’s family planning policy [15,16,17], cost-benefit approaches [18] and input-out measures [19]. Surprisingly, there is little research on whether the post-2013 organizational mergers reallocated funding and expenditure to family planning work, reduced overlapping work practices between departments, eliminated wasteful funding and streamlined information flows and decision-making. One measure of these organizational outcomes are the level of funding and expenditures to each of these aims, which also provides new insights into the quality and quantity of family planning services and population control measures. In particular, funding is especially important in assessing how organizational integration met China’s family planning targets. We collected and analyzed family planning funding and expenditure data in Shandong province, to provide a novel approach to assessing the key outcomes under China’s two-child policy and new organizational arrangements.

## 2. Materials and Methods 

### 2.1. Data Sources

To assess the changes in funding and expenditures, we collected and integrated a range of financial information from two key databases.

#### 2.1.1. Shandong Provincial Health and Family Planning Commission Information System

From the annual reports by Shandong Provincial Health and Family Planning Commission, data were collected on the number of basic public services and family planning funds and expenditure at all levels of government in Shandong province from 2011 to 2016. 

#### 2.1.2. Shandong Provincial Bureau of Statistics

2011–2016 population data and provincial GDP were collected from Shandong Provincial Bureau of Statistics website and Shandong Statistical Yearbook.

#### 2.1.3. Interview Data

In 11 cities in Shandong province in December 2016, qualitative interviews were conducted with 12 family planning managers responsible for overall management work and financial related work of the Health and Family Planning Commission. The aim was to gain more detailed information about the changes and impact of the merger on resource allocations, work organization, overlapping practices and information flows from frontline managers. Specifically, we collected information about changes in financial investment and work priorities after institutional integration, institutional integration progress in different areas and suggestions for family planning work. 

### 2.2. Definitions

#### 2.2.1. Funding of Family Planning Institutions

Each level of government allocated family planning funds to the HFPCs for their work and management. Government funding accounted for 92% to 99% of the total funds each year. For each Health and Family Planning Commission, funds came from the financial allocation by its upper level Health and Family Planning Commission and by the government at its level. For example, the City Health and Family Planning Commission received funds from the Provincial Health and Family Planning Commission and from the City government.

#### 2.2.2. Expenditure of Family Planning Institutions

Family planning expenditures are the spending at each Health and Family Planning Commission level for all work aspects. Family planning expenditure was divided into four parts: (1) Interest-oriented expenditure was the subsidy from the government to families implementing family planning policies, compensating families who faced a greater risk of pension problems and children with a disability, to ensure these families received proper assistance. Interest-oriented expenditure included family planning incentives, support and security spending according to the laws, regulations and policies; (2) public service expenditure mainly includes the fees of family planning for resident population, reproductive health, prenatal and postnatal care, free technical service and so on; (3) capacity-building expenditure improved the capacity of family planning staff and supported infrastructure development of family planning organizations, and enhanced the working network and information communications within and between related departments. Capacity-building expenditure included population and family planning services equipment operation and maintenance, information construction, personnel training and other expenses; (4) management operations expenditure included staffing and public funding for population and family planning management services at different levels, subsidies for family planning staff employed by village or community level and other expenses.

#### 2.2.3. The Social Compensation Fee

The social compensation fee is the financial penalty on families with more than one child, which compensates the government for its investment in social public services. The social compensation fee was collected by the County Health and Family Planning Commission and submitted to the county finance department.

### 2.3. Statistic Analysis

Descriptive statistics and the depiction of changing trends were used to show the financial funding and expenses allocations of family planning work in Shandong province. Microsoft Office Excel 2016 software (Microsoft, State of Washington, USA) was used for data summary and analysis. 

## 3. Results

To assess whether the new family planning organizations implemented management structures and funding levels to promote the two-child policy, we measured the reallocation of funding and expenditure to family planning work, reductions in overlapping work practices between departments, elimination of wasteful funding and streamlining of information flows and decision-making. 

### 3.1. Family Planning Funding

As shown in Table 1, 2013 was a turning point in family planning funding. Before 2013, family planning funding rose from RMB 5.90 billion in 2011 to RMB 8.47 billion in 2013, and then declined to RMB 5.30 billion in 2016. Per capita family planning funds showed a similar trend, reaching its maximum of RMB 87.02 in 2013 before declining to RMB 53.28 in 2016. The share of family planning funds in Shandong total provincial financial expenditures fell each year, halving from 1.27% in 2013 to 0.61% in 2016.

Table 2 reports the funding proportions from each government level in the total family planning budget, which allows the main sources of the funds and their fiscal burden to be analyzed. As shown in Table 2, there was a shift across funding levels. Funds from the national financial transfer payments ceased from 2013. The end of national family planning funds reflected organizational change, where the family planning investment from the national government forms part of the Health and Family Planning Commission budget allocation, rather than calculated separately as a family planning allocation and a health allocation. So we expected to see this mainly accounting reduction in family planning funding, as shown in Table 2, due to the organizational merger. 

Over the period, Table 2 shows that provincial, city and county funding rose, while township funding declined. With the main sources of family planning funds collected at the county and township level, accounting for over 80% of the funds each year, the significant shift in funding was the rise of country funding and the decline in township funding. This reorganization of family planning funding away from the national and township sources towards the county-level sources was the first major financial outcome arising from the organizational integration of HFPCs.

### 3.2. Family Planning Expenditure

Family planning expenditure reflected the allocation of family planning funds by the HFPCs. By analyzing the structure of family planning expenditure we can assess whether the aim of reallocating funding and expenditure to family planning work, reducing overlapping work practices between departments and eliminating wasteful funding was an outcome of the organizational changes to family planning. In Table 3, two major outcomes of the organizational merger were the decline in capacity-building and management operations expenditure. Management operations, accounting for 48% of total expenditures in 2011, was the main expenditure item pre-2013, but declined significantly to 30% in 2016 (see Table 3). Interest-oriented and public service expenditure generally increased, and given the decline in total family planning expenditure, their proportion in total expenses rose year by year. Capacity building and infrastructure management spending rose before 2014, then declined rapidly, accounting for only 5.80% of whole family planning expenditure in 2016.

### 3.3. Social Compensation Fee

The social compensation fee was tied to the number of children born in violation of one-child regulations. As shown in Figure 1, under the new two-child policy the fines collected by the County Family Planning Commissions rose before 2014, then rapidly declined.

### 3.4. Interview Results

Face-to-face interviews with family planning managers in each city reported that more resources were made for family planning work after the organizational merger. The interview results revealed that administrative work procedures were reduced and less time was spent on administration and family size control, with more time allocated to providing family planning services. Similarly, staff were transferred from infrastructure tasks to family planning activities. Importantly, information systems were unified between health and family planning, with more timely information communication and less overlapping activities, which had resulted in a waste of personnel, material and financial resources, and a reduction in work efficiency.

## 4. Discussion

We provided a new assessing method for whether China’s new two-child family planning organizations implemented management structures and funding levels to promote the new family planning policy? By using financial data, we mapped the changes in funds invested in family planning, and their expenditure, pre and post the 2013 establishment of HFPCs. Although total family planning funding fell after the organizational change in 2013, the new HFPCs redirected expenditure towards family planning services. This was achieved by reorganizing the government level funding sources, with increased inputs at the county-level. National and township funding allocations declined. But, without a reduction in family planning services and over 80% of all funding contributed at the county and township levels, these changes suggest that grass-roots organizations mainly carried out family planning work. The delineation of planning and policy work at Provincial and City Health and Family Planning Commission levels and the enhanced family planning service work at County and Township Health and Family Planning Commission levels achieved the organizational merger’s main goal of separating planning and oversight from service provision. The interview evidence also indicated that the organizational merger cut bureaucracy, overlapping work practices and inefficiencies. Better organizational work practices were confirmed by the interview data, with less overlapping practices, better information flows and reduced bureaucratic processes and more staff time allocated to providing front-line family planning services.

While the financial restructurings promoted the two-child policy aims in Shandong province, different county and township local development conditions might challenge the capacity of county and township family planning programs in differ parts of China, especially for underdeveloped and more rural counties and cities, to achieve similar outcomes. In these poorer areas, organizational change may easily lead to insufficient funding and expenditure constraints on family planning work [20,21]. Thus, we recommend that national, provincial and city level governments should ensure that county-level government has the capacity to carry out family planning work.

From the structure of family planning expenditures, we found that management and operation expenditure fell, interest-oriented expenditure remained mainly steady, and public service expenditure rose. These results show the shifting focus of family planning activities in Shandong province. The merger of the family planning commissions and the health commissions in 2013 saw the grass-roots family planning work shift from administrative to service-oriented tasks. The main work content of the merged HFPCs changed from “controlling the population” to “serving the masses and promoting reproductive health” [5]. Before the organizational change, organization management and assessment work occupied a large proportion of the family planning budget. Routine assessment and administrative work were a major component of job execution, and, as a result, family planning staff laid too much stress on administrative work. After the reorganization, the HFPCs better integrated information and resources hastening the transfer of family planning work from management activities to providing services to people in child-bearing ages, such as family planning counseling, eugenics promotion education and distributing contraceptives. 

In contrast to the changes in funding and expenditure, the reaction to the social compensation fee was rapid. No family paid the social compensation fee for a second child, including mothers pregnant at the time of the policy announcement [22]. Few families in China had more than two children. There were few social compensation fees to collect after 2015. 

## 5. Conclusions

To assess whether family planning organizations’ management structures and funding levels and streams shift in response to the new two-child policy and whether the new management structures promoted the new family planning policy, we investigated the family planning funding and expenditure trends before and after the 2013 formation of the National Health and Family Planning Commission, and measured the reallocation of funding and expenditure to family planning work, reductions in overlapping work practices between departments, elimination of wasteful funding and streamlining of information flows and decision-making. We found that the organizational change reduced total family planning funding and saved resources, while increasing family planning services. Planning and oversight at province and city levels were separated from service provision at the county and township levels. After the organizational merger, the main sources of funding came from grass-roots county and township level family planning commissions, and the main work of family planning was undertaken by those same commissions. In line with the new two-child policy, HFPCs turned their attention away from operational management to service provision. At the same time, the reduction of social compensation fee eased people’s financial burden of bearing a second child and guarantee their equal reproductive rights. Overlapping work practices, wasteful funding and information flows and streamlined decision-making were improved. The organizational changes embodied in China’s HFPCs provided a useful template for other countries managing family planning, health services and population control. During a major policy shift from a one-child to a two-child policy, China created new organizations that reduced waste, overlap and bureaucracy, while increasing family planning services. China’s model of successful organizational change provides useful lessons to population planners. HFPCs implemented new management structures and funding levels and streams that promoted the new family planning policy. 

## Figures and Tables

**Figure 1 ijerph-16-02121-f001:**
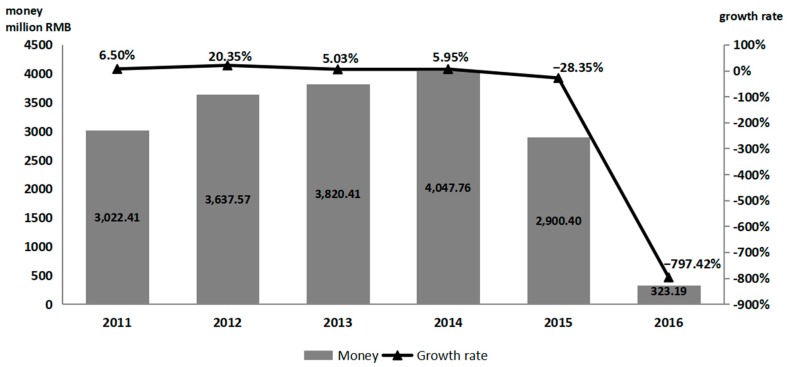
2011–2016 Social Compensation Fee of Shandong Province.

**Table 1 ijerph-16-02121-t001:** 2006–2015 Family Planning Investment Funds, Financial Expenditure and its Proportion in Provincial Financial Expenditure.

Year	Family Planning Investment Funds (RMB billion) (annual growth rate)	Total Provincial Financial Expenditure (RMB billion) (annual growth rate)	Proportion of Family Planning Funds in Financial Expenditure (%)	Permanent Residents (Million)	Per Capita Family Planning Funds (RMB)
2011	5.90	500.21	1.18	96.37	61.22
2012	6.99 (18.47%)	590.45 (18.04%)	1.18	96.85	72.17
2013	8.47 (21.17%)	668.88 (13.28%)	1.27	97.33	87.02
2014	7.40 (−12.63%)	717.73 (7.30%)	1.03	97.89	75.60
2015	6.79 (−8.24%)	825.00 (14.95%)	0.82	98.47	68.96
2016	5.30 (−11.50%)	875.52 (6.12%)	0.61	99.47	53.28

**Table 2 ijerph-16-02121-t002:** Different investment levels of family planning investment in Shandong province (RMB million) (Percentage in total investment).

Year	National Level	Provincial Level	City Level	County Level	Township Level
2011	126.92 (2.34%)	220.02 (4.05%)	366.98 (6.76%)	2076.91 (38.23%)	2641.61 (48.63%)
2012	203.46 (3.07%)	305.26 (4.60%)	485.57 (7.32%)	2683.94 (40.46%)	2955.07 (44.55%)
2013	0	364.05 (4.53%)	742.02 (9.23%)	3779.57 (47.00%)	3155.66 (39.24%)
2014	0	97.34 (1.27%)	562.05 (7.34%)	4426.18 (57.78%)	2575.32 (33.62%)
2015	0	456.53 (7.55%)	542.15 (8.96%)	3453.77 (57.08%)	1597.79 (26.41%)

**Table 3 ijerph-16-02121-t003:** 2011–2015 Classification and Structure of Family Planning Expenditure in Shandong Province.

Year	Total (RMB million)	Interest-Oriented		Public Service		Capacity-Building		Management Operation	
		Amount (RMB million)	Percent	Amount (RMB million)	Percent	Amount (RMB million)	Percent	Amount (RMB million)	Percent
2011	5650.98	1049.04	18.56%	886.90	15.69%	1009.04	17.86%	2706.00	47.89%
2012	6655.97	1413.01	21.23%	1223.72	18.39%	1108.26	16.65%	2910.97	43.73%
2013	8435.33	2070.99	24.55%	1464.03	17.36%	1280.92	15.19%	3619.40	42.91%
2014	8137.45	2273.95	27.94%	1645.27	20.22%	1497.35	18.40%	2720.88	33.44%
2015	7360.31	2644.65	35.93%	1505.45	20.45%	894.54	12.15%	2315.67	31.46%
2016	6036.20	2385.82	39.53%	1486.29	24.62%	349.94	5.80%	1814.14	30.05%

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
