# Peer review of "Evaluating Family Planning Organizations Under China’s Two-Child Policy in Shandong Province"

_ijerph, 2019, doi:10.3390/ijerph16122121_

Round 1

Reviewer 1 Report

 This revision of the manuscript makes it stronger, but I still have some reservations about it.

The authors have still not demonstrated how the new administrative structure is “fit for purpose” – it isn’t clear why that term is needed in the paper.  The “purpose” of the new policy is to get Chinese couples to have more children – that isn’t happening, so how is the new structure “fit for purpose?”  With that said, the paper is useful in showing how funding for the new structure has shifted, but is still adequate for funding family planning services in the country.  Could the authors get rid of the term “fit for purpose” and say that the purpose of the paper is to show how the management structure and funding levels and streams have shifted with the shift in policy and the reaction to the new management structure in terms of ability to carry out tasks under the new family planning policy? 

Some specific comments:

Line 46.  It is confusing to start with talking about “Health and Family Planning Commissions” – what does the plural stand for?  Explain that they are operate at different administrative levels (rather than that there are competing “health” and “family planning” commissions).

Line 91.  What is the evidence that the one child policy led to gender equality?  The highly unbalanced sex ratio that is plaguing China now as men can’t find wives shows that gender equality has not been achieved.

Lines 118-119. How do shifts in funding provide insights into the quality of family planning services? 

Line 155.  “Profit-oriented” doesn’t translate well into English – what is the “profit” part?  Profit to who? 

Line 292 – what is the evidence that family planning services “were enhanced?”  What does “enhanced” mean? 

Author Response

1. Line 46.  Could the authors get rid of the term “fit for purpose” and say that the purpose of the paper is to show how the management structure and funding levels and streams have shifted with the shift in policy and the reaction to the new management structure in terms of ability to carry out tasks under the new family planning policy? 

We have followed the advice of Reviewer 1 and deleted the term “fit for purpose”. We have made changes in the title, Abstract (line 26-28 and 36-37) and on page 2, lines 51-53; page 3, lines 19-20; page 5, line 190-191; page 8, lines 256-257; page 9, lines 300-302 and 322-323.

2. Line 91.  What is the evidence that the one child policy led to gender equality?  The highly unbalanced sex ratio that is plaguing China now as men can’t find wives shows that gender equality has not been achieved.

We have re-written page 3 lines 100-101 on “gender equality”. We note the impact of the one-child policy on the sex ratio, but also reference new research (Ming-Hsuan Le. The One-Child Policy and Gender Equality in Education in China: Evidence from Household Data. Journal of Family and Economic Issues March 2012, Volume 33, Issue 1, pp 41–52) and Yang Hu & Xuezhu Shi. The impact of China’s one-child policy on intergenerational and gender relations, Contemporary Social Science, 2018: 1-18. DOI: 10.1080/21582041.2018.1448941 that suggest more resources were allocated to girls in single child families compared to multi-child families.

3. Lines 118-119. How do shifts in funding provide insights into the quality of family planning services? 

We have deleted the word “quality” on page 3 line 129.

4. Line 155.  “Profit-oriented” doesn’t translate well into English – what is the “profit” part?  Profit to who? 

Page 4 line 167– The previous “Profit-oriented expenditure” was translated from the Chinese term “利益导向类支出”, an expenditure of a type of family planning policy. After consulting managers of family planning work in Shandong province and referring to published articles, a more accurate translation of this policy should be “Interest-oriented policy”, which subsidized those families who implemented family planning policies according to the family planning laws and regulations.

We have revised the translation of the Chinese term “利益导向类支出” to “Interest-oriented expenditure”, rather than “Profit-oriented expenditure”. We have used the new terms on Page 4 line 167 and line 169; page 7 line 234; Table 3; page 9 line 284.

5. Line 292 – what is the evidence that family planning services “were enhanced ? ”  What does “enhanced” mean?

We have replaced “enhanced” to “increased” on page 9, line 309 to make clear that our interview data (Page 8 section 3.4) provided evidence that the resource allocations to family planning services increased the services.

Reviewer 2 Report

Line 31 suggest "financing was reduced" Line 32 suggest "expenditures were shifted" line 33 suggest deleting grass work or use "grass roots" instead line 35 suggest "flows were improved" lines 37-38 suggest "and expenditures on family planning work were enhanced and made more efficient". line 47 suggest "fit for purpose" or working according to their designed objectives?" line 51 suggest "rules of the game" line 104 suggest "Commission was merged into" line 110 suggest "the organizational changes"

Author Response

1. Line 31 suggest "financing was reduced"

Change made page 1 line 32

2. Line 32 suggest "expenditures were shifted"

Change made page 1 line 32

3. line 33 suggest deleting grass work or use "grass roots"

Change made page 1 line 33

4. instead line 35 suggest "flows were improved"

Change made page 1 line 35

5. lines 37-38 suggest "and expenditures on family planning work were enhanced and

made more efficient".

Change made page 1 lines 38-39

6. line 47 suggest "fit for purpose" or working according to their designed objectives?"

In response to Reviewer 1 “fit for purpose” has been deleted and Page 1 lines 53-55 has been re-written: “This paper asks the question: did the management structure and funding levels and streams shift in response to the new two-child policy and did the new management structure promote the new family planning policy?”

7. line 51 suggest "rules of the game"

Change made on page 2 line 59.

8. line 104 suggest "Commission was merged into"

Change made page 3 line 116

9. line 110 suggest "the organizational changes"

Change made page 3 line 118

Round 2

Reviewer 1 Report

Thank you for the final revisions to your paper.  It is a nice contribution to the literature on the evolution of China's family planning policy and implementation.  

This manuscript is a resubmission of an earlier submission. The following is a list of the peer review reports and author responses from that submission.

Round 1

Reviewer 1 Report

This paper addresses on important topic – whether the current organizational structure in China is suited (“fit for purpose”) for implementing China’s new two-child policy than the previous organization.  This paper, which focuses only on the funding under the old and new organizational structures, is not able to answer that question.  Efficacy goes beyond funding – and sources of funding.  The paper suggests that funding for family planning went down with the two child policy – how does that suggest efficacy?  The paper suggests that since funding levels are going down, the program is more effective – but the paper doesn’t provide any evidence of efficacy beyond the funding decline. 

Furthermore, the paper suggests that merging institutions makes them more efficient (or promotes efficacy) but the paper does not say what institutions were merged – health and family planning?  Or was it that the institution that managed family planning in China was extended from the national to lower levels? 

Other comments:

The authors say that the 2015 two-child policy is the most important institutional change in the 60-year history of China’s family planning.  How can the authors ignore the imposition of the One Child policy as the most significant institutional change in family planning in China?  That was in 1979 – 40 years ago – the most significant point in family planning in China – moving to a two child policy pales in comparison to that watershed policy by China’s government.  

The paper covers the history of family planning in China from 1955 to 1978 (lines 58-67) without providing any references.  References are needed. 

Why was the shift made from the Population and FP Commission to the Health and FP Commissions?  Was this a change in name only – and from a national commission to commissions at various levels (e.g. decentralization at national, provincial, city, county and township)?  If not, what aspects of health were included in the new institutional structure, beyond family planning? 

How has the paper shown changes in outcomes – quality and quantity of FP services and health services and population management?  No statistics on these are provided in the paper – the reader is expected to intuit these outcomes from changes in funding levels. 

I have not read anything to suggest that Chinese couples are responding to the call to have two children. How are the authors defining “fit for purpose” if the new policy is not succeeding in increasing the fertility rate? 

The paper mentions a “separate child policy” – what is that?   The One-Child policy has long said that couples in which one of them is an only child could have two children – that isn’t new. 

The funding/expenditure categories noted in the paper are not intuitive for the reader – “profit-oriented”, “public service”, “capacity-building”, and “management operations” – please rethink these categories and make them clearer for readers.  The least clear is the “profit-oriented” category – whey is the family planning program trying to turn a profit?  Would that be the fees charged on families that had more than one child?  What does that mean in the context of the two child policy?    

Reviewer 2 Report

this is an important subject. I was happy to read it.  I would like to have more details on the methodology used.

Detail information on capacity building and management operations expenditure should be included.  what type of capacity building was done?

Authors claim that organizational change reduced total FP funding and saved resources, while enhancing FP services. There is no evidence to prove this concept. what data and evidence was collected to show that FP services were enhanced?  what does FP services comprise of?

Need to add more empirical data to link funding, expenditure and program.